# Effect of Spherical Adsorptive Carbon Among Chronic Kidney Disease Patients: A Nationwide Cohort Study

**DOI:** 10.3390/ijerph22091365

**Published:** 2025-08-30

**Authors:** Dong Hui Shin, Keunryul Park, Jae Won Yang, Jun Young Lee

**Affiliations:** Department of Nephrology, Comprehensive Kidney Disease Research Institute, Wonju College of Medicine, Yonsei University, Wonju 26426, Republic of Korea; dhshin@yonsei.ac.kr (D.H.S.); hkspy.park@gmail.com (K.P.); kidney74@yonsei.ac.kr (J.W.Y.)

**Keywords:** cohort studies, disease progression, kidney diseases, chronic, renal replacement therapy, survival analysis

## Abstract

Spherical Adsorptive Carbon (SAC), a type of oral sorbent, is prescribed to chronic kidney disease (CKD) patients to remove uremic toxins. However, evidence regarding its effectiveness in delaying chronic kidney disease (CKD) progression remains insufficient. We aimed to evaluate the impact of SAC on CKD progression in patients with CKD stage 3 or higher using nationwide data. In this retrospective cohort study, we included patients diagnosed with CKD stage ≥3 from the Korea National Health Insurance System database between January 2020 and December 2022. Outcomes were compared between SAC users (N = 1289) and non-users (N = 1289) after 1:1 propensity score matching (PSM). After PSM, the time from index date to end-stage kidney disease (ESKD) was significantly longer in the SAC user group compared to the non-user group (246.8 days vs. 118.6 days, *p* < 0.001). In Cox regression analysis, the risk of ESKD was significantly lower in the SAC group (HR = 0.37, 95% CI: 0.29–0.48). However, the risk of dialysis initiation did not show a significant difference between the two groups (HR = 0.83, 95% CI: 0.27–2.59). This nationwide cohort study suggests that SAC treatment may delay progression from CKD stage 3 to ESKD, although it did not significantly reduce the risk of dialysis initiation.

## 1. Introduction

The prevalence of chronic kidney disease (CKD) has been steadily increasing, affecting approximately 10% of the population [1,2]. In particular, the incidence of end-stage kidney disease (ESKD) has risen dramatically [2,3]. Recently the development of sodium glucose co-transporter 2 (SGLT2) inhibitors and glucagon like protein (GLP)-1 agonists has provided new therapeutic options to slow the progression of CKD [4]. However, their effectiveness in delaying dialysis initiation in patients with CKD stage 4 or higher remains unclear or appears to be restricted to diabetic patients. Treatment options for delaying dialysis initiation in patients with a GFR below 30 mL/min/1.73 m^2^ are quite limited [5].

Spherical Adsorptive Carbon (SAC) has emerged as a potential treatment to slow CKD progression through a different mechanism [6]. SAC attenuates CKD progression by adsorbing uremic toxins and their precursors, such as indoxyl sulfate (IS) and indole, from the gastrointestinal tract [7,8,9]. Indole derived from the metabolism of tryptophan by gut bacteria serves as a precursor of IS [9,10]. Moreover, SAC may suppress oxidative stress and reduce cardiovascular damage [11,12].

Although randomized controlled trials (RCTs) have been conducted to evaluate the efficacy of SAC in patients with CKD, real-world effectiveness has not yet been assessed using national cohort data [6,8,10,11,13]. Therefore, we aim to analyze the real-world effectiveness of SAC using national health insurance data.

## 2. Materials and Methods

The data were selected from the anonymized Health Insurance Review and Assessment Service (HIRA) database, which assesses medical services and maintains quality standards by reviewing healthcare claims. The database includes patient demographics and detailed diagnostic data [14]. Medical providers in Korea are required to submit inpatient and outpatient claims to HIRA for reimbursement of medical procedures covered by the Korean National Health Insurance System. Since 97% of the Korean population is enrolled in health insurance, the HIRA database is representative of the entire population. As this study is a retrospective analysis of the nationwide claims database, no formal a priori sample size calculation is performed. Instead, all eligible patients were included in the cohort, which provided adequate statistical power for the analyses.

This nationwide, population-based retrospective cohort study aimed to compare the time to ESKD onset and the risk of ESKD between patients using SAC and those not using SAC. Furthermore, we compared the risk of dialysis initiation between the two groups.

### 2.1. Study Population

We initially identified 202,286 patients from the HIRA database who were diagnosed with CKD stage ≥3 between 2020 and 2022 in Korea. We then retrospectively extracted their data from 2018 to 2023. CKD stage ≥3 was operationally defined using the International Classification of Diseases, 10th Revision (ICD-10) codes N18.3, N18.4, and N18.5. Patients younger than 18 years old or above 80 years (n = 44,373), diagnosed with cancer, kidney transplantation (KT), CKD stage 4, or dialysis within one year before the first diagnosis of CKD stage 3 (n = 67,971), prescribed spherical adsorptive carbon before the first diagnosis of CKD stage 3 (n = 1771), who first diagnosed CKD stage 3 after 2022 (n = 33,000), who did not prescribe the SAC continuously for 90 days or more (n = 1078), who lost follow up (n = 40,672), who expired at CKD stage 3 diagnosis (n = 23), whose CKD stage 3 diagnosis date was the last visit date (n = 23), and a follow up date below seven days (n = 429) were excluded. These exclusions correspond to cases with missing or insufficient data for defining exposures or outcomes (e.g., early death, short follow-up), and imputation was not feasible in this claims-based dataset. After exclusion there were two kinds of patients who prescribed SAC (SAC group, n = 1082) and did not prescribe SAC (non-user group, n = 11,864). SAC exposure was defined based on the Health Insurance Review & Assessment (HIRA) service code ‘4597’ (Figure 1).

### 2.2. Propensity Score Matching

To reduce immortal-time bias in the comparison between the SAC group and the non-user group, we randomly assigned the index date for the non-user group, while the SAC group was assigned the index date based on the SAC initiation date. Additionally, we matched the duration from CKD stage 3 to the index date, the year of the index date, and the quarter of CKD stage 3 diagnosis.

Subsequently, we performed propensity score matching (PSM) in a 1:1 ratio using the greedy (nearest neighbor) matching between the SAC group and non-user group. The propensity score, which represents the probability of receiving SAC, was estimated using a logistic regression model. The model included baseline covariates such as age, sex, NSAID use, steroid use, ACE inhibitor or ARB use, cyclosporine use, SGLT2 inhibitor use, Charlson Comorbidity Index (CCI), hypertension (HTN), diabetes mellitus (DM), and dyslipidemia. After PSM, we evaluated covariate balance by calculating the standardized mean difference (SMD) between groups, with a value less than 0.1 considered acceptable. Details on the codes and medications used to define each diagnosis, procedure, and drug in this study are shown in Appendix A.

### 2.3. Outcomes

The primary outcome of this study was the time to ESKD, and the secondary outcomes were the initiation of dialysis and all-cause death. In addition, we planned a subgroup analysis stratified by the duration of drug use to explore whether long-term exposure to the SAC influences its efficacy. We operationally defined these outcomes based on ICD-10 codes, which were recorded as the primary and secondary diagnosis by a physician in either an inpatient or outpatient setting. In this study, we defined ESKD as who received dialysis (peritoneal dialysis, hemodialysis, or kidney transplantation or combination of those treatments). Therefore, ESKD was defined by codes N18.5 or N18.6, and initiation of dialysis was defined by codes Z49.1 or Z49.2 (which requires at least 3 months of maintenance hemodialysis or peritoneal dialysis or received immunosuppressive agents). The follow-up period was calculated from the index date to the date of the first outcome, the date of their last clinic visit, or the end of study period (31 December 2023), whichever came first. Subjects with a follow-up duration of less than 7 days or an average annual visit frequency of less than 6 times were excluded from the study based on exclusion criteria.

### 2.4. Statistical Analysis

Baseline characteristics between the SAC and non-user groups, both before and after PSM, were compared using standardized mean difference (SMD). Categorical and continuous variables are presented as numbers, percentages, means, and standard deviations. In the PS-matched cohort, we used Kaplan–Meier survival curves and the log-rank test to compare the cumulative incidence of all events between the SAC and non-user groups. We compared the time to ESKD from index date between groups, with ESKD defined as an event of interest. Hazard ratios (HRs) for incidence of ESKD and dialysis were determined after PSM using univariate Cox regression models. Additionally, we analyzed the HRs for initiation of dialysis according to the total duration of drug administration (<180 days, 180–365 days, and >365 days). No specific handling for missing values was performed, and the raw data from the HIRA database were used for all analyses. All statistical analyses were performed using SAS Enterprise Guide, version 7 (SAS Institute Inc., Cary, NC, USA), and a *p*-value < 0.05 was considered statistically significant.

## 3. Results

### 3.1. Baseline Characteristics

The baseline characteristics of unmatched and matched patients are presented in Table 1. Before PSM, the SAC group had lower mean age, higher ACEi or ARB user, HTN patients, and DM patients. After PSM, all covariates were well balanced between the SAC user group (n = 1082) and the non-user group (n = 1082). As shown in Table 1, the standardized mean difference (SMD) was <0.1 for all covariates, indicating adequate balance between the matched groups. Detailed baseline characteristics of both groups before and after PSM are provided in Table 1. The median follow-up duration was 402 (interquartile range (IQR), 253–452) days for the SAC group and 303 (IQR, 125–432) days for the non-user group.

### 3.2. Comparison of Clinical Outcomes Between the SAC and Non-User Groups

When reviewing the Kapan–Meier curves and performing the log-rank test for all events, the SAC user group demonstrated a significantly lower event rate (*p* < 0.001).

The duration from index date to the ESKD was significantly different between the two groups (246.8 days vs. 118.6 days, *p* < 0.001). Consistently, the risk of ESKD (HR = 0.37, 95% CI: 0.29–0.48) was significantly lower in the SAC group than in the non-user group. The median delay in progression to ESKD was approximately 128 days (Table 2). In addition, the risk of death (HR = 0.32, 95% CI: 0.21–0.48) was significantly lower in the SAC group than in the non-user group (Table 3 and Figure 2).

In Cox regression analysis, the risk of dialysis was not significantly different between the SAC group and the non-user group (HR = 0.83, 95% CI: 0.27–2.59) (Table 3). When the results were stratified by the duration of SAC use, the HR showed a decreasing trend with longer medication prescription duration. In patients who took the medication for more than 365 days the observed HR was decreased (HR = 0.25, 95% CI: 0.08–0.87) (Table 4).

## 4. Discussion

In this nationwide cohort study conducted in South Korea, among patients with CKD, 1082 (8.4%) received SAC treatment between 2020 and 2022. After PSM analysis comparing these patients with 1289 SAC naïve 1082 CKD patients, SAC treatment was associated with a significantly lower risk of death and delayed progression of CKD (stage 3 or higher) to ESKD.

SAC exerts its beneficial effects on CKD primarily by reducing intestinal absorption of uremic toxins, especially indoxyl sulfate (IS), thereby attenuating the progression of renal impairment. IS promotes CKD progression by inducing renal fibrosis, oxidative stress, and inflammation, impairing antioxidant systems in renal tubular and glomerular mesangial cells, and downregulating renal proximal tubule expression of klotho in animal models [15,16,17,18,19]. Consistent with this, dietary strategies that reduce uremic toxins, such as low protein diets, have also demonstrated reduced serum IS levels and delayed CKD progression in uremic rat models [20]. Several animal studies demonstrated that SAC effectively decreased urinary IS levels, ameliorated renal dysfunction, increased renal klotho expression, and mitigated pathological changes such as glomerular hypertrophy, glomerulosclerosis, interstitial fibrosis, and proteinuria [17,18,21,22,23,24]. In clinical studies SAC treatment reduced serum and urinary IS levels, as well as advanced glycation end products (AGEs), which are known to accelerate nephropathy by enhancing fibrogenic and inflammatory cytokines expression [21,25].

Several studies have reported that SAC provides cardiovascular benefits for patients with CKD. In the CKD rat model, SAC reduced oxidative stress, which is closely linked to cardiac hypertrophy and fibrosis [26]. Additionally, in vivo study demonstrated that SAC inhibits monocyte activation, a crucial factor in the progression of atherosclerosis [27]. Consistent with these experimental findings, clinical studies involving CKD patients showed that SAC treatment significantly reduced carotid intima-media thickness (IMT) and pulse wave velocity, both surrogate markers of cardiovascular risk [28]. Furthermore, a retrospective cohort study from Japan reported that SAC treatment was associated with a reduction in cardiovascular events among patients with CKD stage 3–5 compared to untreated CKD stage 3–5 patients [29].

Despite these promising findings, the clinical effectiveness of SAC remains controversial. Some retrospective cohort studies reported a reduction in mortality risk, similar to our findings [29,30,31]. However, other retrospective studies and large randomized controlled trials failed to demonstrate a significant association between SAC use and mortality [10,13,32,33,34]. In particular, the multinational phase III trials (EPPIC-1 and EPPIC-2) did not show a reduction in progression to ESKD with SAC compared to placebo and a Japanese multicenter RCT by Akizawa et al., who similarly reported no clear mortality or renal benefit [33,34]. There neutral or negative findings highlight the ongoing uncertainty and suggest that the effectiveness of SAC may vary by patient population and clinical context.

The controversy regarding the therapeutic effectiveness of SAC may be attributed to the following factors. First, most frequent limitation of SAC treatment is the need of a high dose of drugs. However, like our study, most of the studies did not report compliance of medication. There has been a report from Japan indicating that approximately 70% of patients experience medication adherence issues with SAC due to problems such as the required dosage amount, the frequency of administration, the necessity of taking it separately from other medications, and gastrointestinal disturbances [35]. Second, patients with CKD may exhibit considerable clinical heterogeneity due to differences in age, underlying etiology, disease severity, and other individual characteristics, which can affect treatment responses. Retrospective studies evaluating the benefits of SAC are often limited by their inability to account fully for all patient-related factors. Therefore, this heterogeneity poses a significant limitation when interpreting and generalizing the effects of SAC reported in observational studies.

In our study, the hazard ratio for ESKD was significantly reduced in the SAC group, whereas the hazard ratio for dialysis initiation did not reach statistical significance. This discrepancy may be explained by differences in outcome definitions and operational characteristics. ESKD was defined as a composite endpoint including dialysis initiation, kidney transplantation, and diagnostic codes, whereas dialysis initiation alone represents a narrower definition and thus a smaller number of events, leading to lower statistical power. In addition, patients who died before starting dialysis were censored in the dialysis analysis, which could attenuate the apparent effect of SAC on this outcome. The initiation of dialysis is also influenced by physician judgment, patient preference, and reimbursement policies, introducing greater variability compared with the more standardized definition of ESKD. Finally, the relatively short follow-up period in some patients may have further contributed to these differences. Together, these factors may explain the observed discrepancy and highlight the need for careful interpretation of dialysis-specific outcomes in claims-based studies.

Our study has several limitations. First, due to the inherent limitations of the claims database, we could not obtain detailed clinical information, such as laboratory results, exact SAC dosages, or primary kidney disease. Consequently, CKD stages could not be determined by creatinine and urine albumin levels, as recommended by the KDIGO guidelines, but were instead identified using ICD codes. In addition, we were unable to evaluate the use of cyclophosphamide or to distinguish the exact type of RAAS blockers, because such information was not available in the claims data. Therefore, selection bias associated with these unmeasured factors could not be entirely excluded. Second, we could not assess actual medication adherence (e.g., proportion of days covered) or long-term treatment persistence. While the data confirm the dispending of SAC prescriptions, they do not indicate whether the medication was taken as prescribed, for how long, or if it was taken continuously without interruption. Therefore, the prescription duration may not accurately reflect the actual treatment exposure. In addition, we performed a subgroup analysis stratified by the duration of drug use to explore whether long-term exposure to SAC influences its efficacy. While these findings provided additional insights, they should be interpreted with caution, as the subgroup analyses were exploratory in nature and no formal adjustment for multiple comparisons was applied. Potential sources of bias, such as immortal time bias or healthy user effects, may also have influenced the results, particularly in the long-duration group.

## 5. Conclusions

In conclusion, SAC treatment was associated with a lower risk of all-cause mortality and a delay in CKD progression, suggesting a potential clinical benefit in patients with stage 3 CKD. While these findings highlight a favorable risk–benefit profile, the retrospective and observational design imposes limitations. Therefore, confirmation through high-quality, large-scale, multicenter prospective randomized controlled trials is essential to establish the role of SAC in CKD management.

## Figures and Tables

**Figure 1 ijerph-22-01365-f001:**
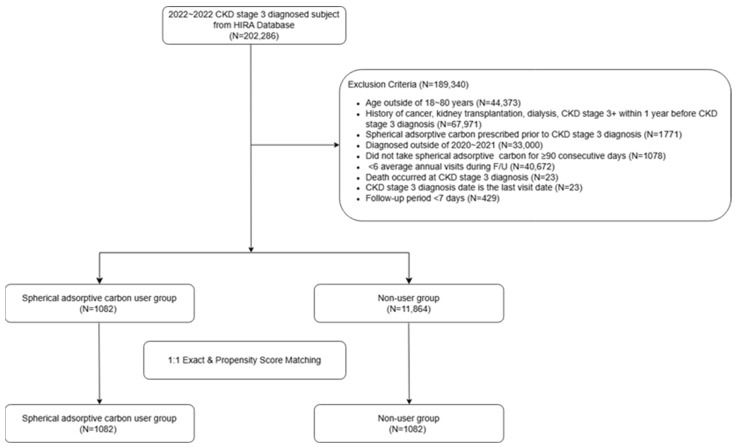
Flow diagram of study patients.

**Figure 2 ijerph-22-01365-f002:**
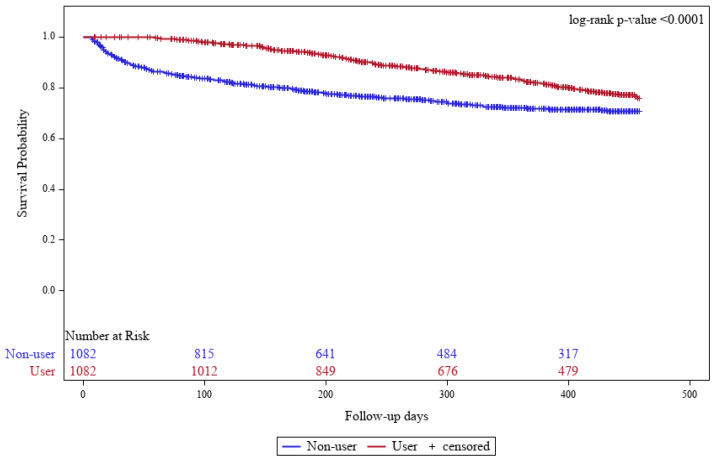
Kaplan–Meier curve for survival.

**Table 1 ijerph-22-01365-t001:** Baseline characteristics of two groups before and after matching.

	Before Matching (N = 12,946)			After Matching (N = 2164)		
	SAC Group	Non-User Group	SMD	SAC Group	Non-User Group	SMD
	(N = 1082)	(N = 11,864)		(N = 1082)	(N = 1082)	
Quarter of CKD stage 3			0.085			0.000
202001	103 (9.52)	1100 (9.27)		103 (9.52)	103 (9.52)	
202002	124 (11.46)	1231 (10.38)		124 (11.46)	124 (11.46)	
202003	132 (12.20)	1401 (11.81)		132 (12.20)	132 (12.20)	
202004	113 (10.44)	1406 (11.85)		113 (10.44)	113 (10.44)	
202101	116 (10.72)	1483 (12.50)		116 (10.72)	116 (10.72)	
202102	156 (14.42)	1556 (13.12)		156 (14.42)	156 (14.42)	
202103	152 (14.05)	1613 (13.60)		152 (14.05)	152 (14.05)	
202104	186 (17.19)	2074 (17.48)		186 (17.19)	186 (17.19)	
Duration from CKD stage 3 to index date			0.810			0.000
Months	2.98 (3.90)	6.38 (4.47)		2.98 (3.90)	2.98 (3.90)	
Year of index date			0.462			0.000
2020	335 (30.96)	2221 (18.72)		335 (30.96)	335 (30.96)	
2021	579 (53.51)	5704 (48.08)		579 (53.51)	579 (53.51)	
2022	167 (15.43)	3769 (31.77)		167 (15.43)	167 (15.43)	
2023	1 (0.09)	170 (1.43)		1 (0.09)	1 (0.09)	
Age, years	63.09 (12.23)	65.85 (11.80)	0.229	63.09 (12.23)	64.11 (12.05)	0.084
Sex, male (%)	827 (76.43)	7595 (64.02)	0.274	827 (76.43)	834 (77.08)	0.015
Prior medication, n (%)						
NSAID	926 (85.58)	10,312 (86.92)	0.039	926 (85.58)	951 (87.89)	0.068
Steroid	733 (67.74)	8204 (69.15)	0.030	733 (67.74)	740 (68.39)	0.014
ACE Inhibitor or ARBs	825 (76.25)	8398 (70.79)	0.124	825 (76.25)	840 (77.63)	0.033
Cyclosporine or Tacrolimus	36 (3.33)	363 (3.06)	0.015	36 (3.33)	31 (2.87)	0.027
SGLT2 Inhibitor	227 (20.98)	2313 (19.5)	0.037	227 (20.98)	230 (21.26)	0.007
CCI Group, n (%)						0.073
0~2	310 (28.65)	3756 (31.66)		310 (28.65)	279 (25.79)	
3~5	492 (45.47)	5129 (43.23)		492 (45.47)	496 (45.84)	
6~8	234 (21.63)	2441 (20.57)		234 (21.63)	258 (23.84)	
≥9	46 (4.25)	538 (4.53)		46 (4.25)	49 (4.53)	
Other comorbidity, n (%)						
Hypertension	1043 (96.40)	11,060 (93.22)	0.143	1043 (96.40)	1048 (96.86)	0.026
Diabetes mellitus	950 (87.80)	9573 (80.59)	0.196	950 (87.80)	973 (89.93)	0.068
Dyslipidemia	1040 (96.12)	11,164 (94.1)	0.094	1040 (96.12)	1048 (96.86)	0.040

Abbreviations: ACE = angiotensin converting enzyme; ARB = angiotensin receptor blocker; CCI = Charlson comorbidity index; CKD = chronic kidney disease; N = number; NSAIDs = non-steroidal anti-inflammatory drugs; SAC = spherical adsorptive carbon; SGLT2i = sodium-glucose cotransporter 2 inhibitor; SMD = standard mean difference.

**Table 2 ijerph-22-01365-t002:** Time to ESKD (days) and hazard ratio of ESKD according to SAC use after matching.

	SAC User Group (N = 1082)	Non-User Group (N = 1082)	*p*-Value
Events, No. (%)	153 (14.14)	115 (10.63)	
Time to events, days			
Mean(SD)	246.8 (106.0)	118.6 (109.2)	<0.0001 *
Q1, Q3	98, 388	18, 118	
Incidence rate, per 100 PY	15.75	15.35	
Crude HR (95% CI)	0.37 (0.29–0.48)	Ref.	<0.0001 **

* Mann–Whitney U test. ** Cox regression model. Abbreviations: PY = person-year; HR = hazard ratio.

**Table 3 ijerph-22-01365-t003:** Number of events, incidence rates, and Cox proportional hazard ratio of dialysis and death after matching.

	SAC User Group	Non-User Group	Crude HR (95% CI)	*p*-Value
	(N = 1082)	(N = 1082)		
Dialysis				
Events, No.(%)	5 (2.65)	8 (2.96)		
Incidence rate, per 100 PY	0.51	1.07	0.83 (0.27–2.59)	0.7519
Death				
Events, No.(%)	31 (16.4)	147 (54.44)		
Incidence rate, per 100 PY	3.19	19.62	0.32 (0.21–0.48)	<0.0001

Abbreviations: PY = person-year; HR = hazard ratio.

**Table 4 ijerph-22-01365-t004:** Hazard ratio of dialysis according to SAC prescription days.

SAC Prescription Days	HR * (95% CI)	*p*-Value
90–179 days	1.50 (0.75–3.43)	0.2237
180–364 days	0.36 (0.11–1.23)	0.1035
365 days	0.25 (0.08–0.87)	0.0285

* Ref: Non-user group.

## Data Availability

HIRA provides data with the approval of HIRA through the Korean Health Insurance Review and Assessment Service (https://www.hira.or.kr/main.do (accessed on 10 June 2025).

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
