# Peer review of "Effect of Spherical Adsorptive Carbon Among Chronic Kidney Disease Patients: A Nationwide Cohort Study"

_ijerph, 2025, doi:10.3390/ijerph22091365_

Round 1

Reviewer 1 Report

Comments and Suggestions for Authors

The study uses a strong national registry-based design with a large sample size and appropriate matching methods to address confounding, which strengthens its external validity. However, there are several areas where clarification or improvement is needed:

Major:

  1. The overall hazard ratio for drug users vs. non-users was not statistically significant, but significant associations were observed in duration-based subgroups. The authors should clarify whether these stratified analyses were pre-specified or exploratory, and whether any adjustment for multiple comparisons was made. Additionally, potential biases such as immortal time bias or healthy user effects in the long-duration group should be acknowledged and discussed.
  2. Please clarify the matching method used (e.g., propensity score matching) and provide details on how covariate balance was assessed post-matching (e.g., target SMD thresholds).
  3. The methods section should clearly define the outcome variables. At present, it appears there may be two primary outcomes, but this is not explicitly stated. A clearer description of primary and secondary outcomes is needed.

Minor:

  1. The methods section mentions statistical tests such as t-tests and ANOVA, yet Table 1 presents standardized mean differences (SMDs) instead of p-values. The authors should clearly explain whether inferential testing or SMDs were used to assess baseline balance and why one approach was chosen over the other.
  2. The terminology for drug users is inconsistent throughout the manuscript. For instance, the abstract refers to "AST120 users", whereas the main text uses "SAC group". Please standardize the terminology across the entire paper to avoid confusion and improve readability.
  3. Tables S3 and S4 contain key results and may be more appropriately placed in the main manuscript
  4. Lines 96–98 should not be part of the statistical analysis subheading. These sentences appear to be general descriptions.

Author Response

Reviewer 1

Comments and Suggestions for Authors

The study uses a strong national registry-based design with a large sample size and appropriate matching methods to address confounding, which strengthens its external validity. However, there are several areas where clarification or improvement is needed:

Major:

  1. The overall hazard ratio for drug users vs. non-users was not statistically significant, but significant associations were observed in duration-based subgroups. The authors should clarify whether these stratified analyses were pre-specified or exploratory, and whether any adjustment for multiple comparisons was made. Additionally, potential biases such as immortal time bias or healthy user effects in the long-duration group should be acknowledged and discussed.

 Answer) Thank you for pointing this out. We updated manuscript as follows: “To reduce immortal-time bias in the comparison between the SAC group and the non-user group, we randomly assigned the index date for the non-user group, while the SAC group was assigned the index date based on the SAC initiation date. Additionally, we matched the duration from CKD stage 3 to the index date, the year of the index date and quarter of CKD stage 3 diagnosis”, “In addition, we planned a subgroup analysis stratified by the duration of drug use to explore whether long-term exposure to the SAC influences its efficacy”.  “In addition, we performed a subgroup analysis stratified by the duration of drug use to explore whether long-term exposure to SAC influences its efficacy. While these findings provided additional insights, they should be interpreted with caution, as the subgroup analyses were exploratory in nature and no formal adjustment for multiple comparisons was applied. Potential sources of bias, such as immortal time bias or healthy user effects, may also have influenced the results, particularly in the long-duration group”, and “We acknowledge that the subgroup analyses according to duration of drug use were exploratory and may be subject to potential biases, including immortal time bias and healthy user effects. Moreover, no adjustment for multiple testing was performed. These limitations should be taken into account when interpreting our findings”.

  1. Please clarify the matching method used (e.g., propensity score matching) and provide details on how covariate balance was assessed post-matching (e.g., target SMD thresholds)

Answer) Thank you for pointing this out. We updated material and method section as follow: “To reduce immortal-time bias in the comparison between the SAC group and the non-user group, we randomly assigned the index date for the non-user group, while the SAC group was assigned the index date based on the SAC initiation date. Additionally, we matched the duration from CKD stage 3 to the index date, the year of the index date and quarter of CKD stage 3 diagnosis. Subsequently, we performed propensity score matching (PSM) in a 1:1 ratio using the greedy (nearest neighbor) matching between the SAC group and non-user group. The propensity score, which represents the probability of receiving SAC, was estimated using a logistic regression model. The model included baseline covariates such as age, sex, NSAID use, steroid use, ACE inhibitor or ARB use, cyclosporine use, SGLT2 inhibitor use, Charlson Comorbidity Index (CCI), hypertension (HTN), diabetes mellitus (DM), and dyslipidemia. After PSM, we evaluated covariate balance by calculating the standardized mean difference (SMD) between groups, with a value less than 0.1 considered acceptable”.

  1. The methods section should clearly define the outcome variables. At present, it appears there may be two primary outcomes, but this is not explicitly stated. A clearer description of primary and secondary outcomes is needed.

 Answer) Thank you for pointing this out. We updated manuscript as follow “The event of this study was defined as the earliest occurrence of one of the following: initiation of dialysis, progression to ESKD, or all-cause death. The primary outcome of this study was the time to ESKD and the secondary outcome was the hazard ratio of the initiation of dialysis”.

Minor:

  1. The methods section mentions statistical tests such as t-tests and ANOVA, yet Table 1 presents standardized mean differences (SMDs) instead of p-values. The authors should clearly explain whether inferential testing or SMDs were used to assess baseline balance and why one approach was chosen over the other.

Answer) We updated methods as follow: “Baseline characteristics between the SAC and non-user groups, both before and after PSM, were compared using standardized mean difference (SMD)”

  1. The terminology for drug users is inconsistent throughout the manuscript. For instance, the abstract refers to "AST120 users", whereas the main text uses "SAC group". Please standardize the terminology across the entire paper to avoid confusion and improve readability.

Answer) Thank you for pointing this out. We updated the terminology across the entire manuscript as “SAC”. 

  1. Tables S3 and S4 contain key results and may be more appropriately placed in the main manuscript

Answer) We replaced Table S3 and S4 to the main manuscript.

  1. Lines 96–98 should not be part of the statistical analysis subheading. These sentences appear to be general descriptions.

Answer) You are right, we changed that to IRB statement section.

Reviewer 2 Report

Comments and Suggestions for Authors

The research is interesting and important and included a significant number of patients with CKC in Korea, but it is necessary to answer the following questions in this study and thus increase its importance!

- Stage 3 must be defined according to KDIGO guidelines!
- Is it 3A or 3B because the length of CKD progression is variable, is Ccr near 60 or 30 ml/min?
- Is it necessary to define the primary kidney disease in both groups because it also affects the progression of CKD?
- Did the patients in both groups receive other important renoprotective drugs?
- Were there any patients who received cyclophosphamide or only Tac or CsA?
In the first place, RAAS blockers should be separated into ACEi and ATR2!
- SGLT2-inhibitors (empa or dapa) were used in recoprotection or for D2M?
- Were there any patients who received predialysis ESA?
- The discussion must be expanded!

Author Response

Reviewer 2

Comments and Suggestions for Authors

The research is interesting and important and included a significant number of patients with CKC in Korea, but it is necessary to answer the following questions in this study and thus increase its importance!

- Stage 3 must be defined according to KDIGO guidelines!

Answer) Thank you for pointing this out. We fully agree that CKD stage 3 ideally defined according to the KDIGO guidelines, using creatinine-based eGFR and urine albumin levels. However, because our study was conducted using the Health Insurance Review and Assessment Service (HIRA) database, laboratory values were not available. Therefore, CKD stage 3 was defined based on ICD codes, which we considered to reflect KDIGO-based diagnostic practice.

We acknowledge this discordance and have added it to the Discussion section as follows:
“Therefore, CKD stages could not be determined by creatinine and urine albumin levels, as recommended by the KDIGO guidelines, but were instead identified using ICD codes (line 230).”

- Is it 3A or 3B because the length of CKD progression is variable, is Ccr near 60 or 30 ml/min?

Answer) Thank you for pointing this out. Because the ICD-10 codes in our dataset could not distinguish between CKD stage 3A and 3B, we were only able to identify patients as having stage 3 CKD overall. In addition, information on creatinine clearance was not available in the database.

- Is it necessary to define the primary kidney disease in both groups because it also affects the progression of CKD?

Answer) Thank you for pointing this out. Unfortunately, the HIRA database does not contain information that would allow us to identify the primary kidney disease for each patient. Therefore, we were not able to distinguish or adjust for the underlying primary renal disease in our analysis. We have acknowledged this limitation in the Discussion section.

- Did the patients in both groups receive other important renoprotective drugs?

Answer) Thank you for this important question. In our analysis, we examined the use of several key renoprotective agents, including corticosteroids, ACE inhibitors or ARBs, cyclosporine, and SGLT2 inhibitors. Because the study period was limited to 2022, we did not observe the introduction of additional renoprotective agents beyond these. Therefore, our evaluation was restricted to the above-mentioned medications.

- Were there any patients who received cyclophosphamide or only Tac or CsA?
In the first place, RAAS blockers should be separated into ACEi and ATR2!

Answer) Thank you for your comment. Regarding immunosuppressive agents, we were able to identify prescriptions for tacrolimus and cyclosporine in the claims data. However, the use of cyclophosphamide could not be reliably ascertained because its reimbursement in Korea is restricted, and therefore prescription records may not fully reflect actual clinical use. With respect to renoprotective agents, we agree that RAAS blockers should ideally be analyzed separately as ACE inhibitors and ARBs. However, in our dataset it was not possible to fully distinguish ACE inhibitors from ARBs, and therefore we analyzed them together as a single category of RAAS blockers. We have clarified this limitation in the revised manuscript.

- SGLT2-inhibitors (empa or dapa) were used in recoprotection or for D2M?

Answer) Thank you for pointing this out. SGLT inhibitor were used in both renoprotection or D2M.

- Were there any patients who received predialysis ESA?

Answer) Thank you for pointing this out. We excluded patients who were already on dialysis. According to the Korean insurance policy, patients can receive ESA therapy at the predialysis stages. Therefore, all patients included in our study could have received ESA before the initiation of dialysis.-

The discussion must be expanded!

Answer) Thank you for pointing this out. We expanded discussion section.

Reviewer 3 Report

Comments and Suggestions for Authors

Dear Authors,

I have reviewed your manuscript with interest, as it addresses a topic of clear relevance to the field of nephrology. I commend you for your work and appreciate your contribution. In order to enhance the scientific rigor and clarity of the article, I suggest a series of minor revisions, many of which are aligned with the Strengthening the Reporting of Observational Studies in Epidemiology (STROBE) Statement guidelines.

Title

Please consider including the study design explicitly in the title, as recommended by STROBE.

Abstract

I recommend rewriting the abstract using a structured format and specifying the exact type of effect measures used.

Keywords

Ensure that the keywords correspond to MeSH terms. I suggest the following:

  • Kidney Diseases, Chronic
  • Charcoal
  • Uremic Toxins
  • Disease Progression
  • Renal Replacement Therapy
  • Survival Analysis
  • Cohort Studies

Introduction

The introduction would benefit from a clearer and more logical narrative structure. I propose the following flow:

  1. Epidemiological burden of chronic kidney disease (CKD) and consequences of progression to end-stage kidney disease (ESKD).
  2. Limitations of current therapies in patients with eGFR <30 ml/min/1.73m² without diabetes.
  3. Concise description of AST-120, its mechanisms of action, and previous evidence (including both positive and negative findings).
  4. Specific knowledge gap: lack of population-based cohort studies evaluating hard outcomes in real-world settings.
  5. Study objective: a clear and direct statement summarizing the aim of the research.

Methods

  • Clearly define the operational criteria for ESKD, specifying whether it includes chronic hemodialysis, peritoneal dialysis, kidney transplantation, or a combination thereof.
  • Define “initiation of dialysis,” including the type, clinical criteria used, and minimum duration required to consider it an event.
  • Include in the main text (not only in supplementary material) the diagnostic and procedural codes used to identify comorbidities and medications, along with the look-back period.
  • Indicate whether immortal time bias and other time-related biases inherent to retrospective designs were assessed and addressed.
  • Discuss potential residual confounding due to missing clinical variables (e.g., baseline eGFR, proteinuria) and how this may affect interpretation.
  • Specify whether adherence to AST-120 was evaluated (e.g., proportion of days covered or treatment persistence). If not available, please state this explicitly as a limitation.
  • Report whether missing data were present and describe the method used to handle them (e.g., exclusion, imputation).
  • Include a sample size calculation or justify its absence, indicating the expected incidence of ESKD and the clinically relevant effect size intended to be detected.
  • Explain the rationale for selecting the 2020–2022 time frame and its suitability for evaluating CKD progression outcomes in stage 3 patients.
  • Consider conducting sensitivity analyses (e.g., excluding patients with extreme comorbidity) and stratified analyses by key variables (age, sex, diabetes status) to explore effect heterogeneity.

Results

  • Clarify whether dialysis initiation is a subcomponent of the ESKD outcome or a distinct definition. The current presentation of a non-significant HR for dialysis and a significant HR for ESKD may be confusing without conceptual and operational clarification.
  • Include a quantitative summary of key results in the main text (number of events, cumulative incidence, adjusted HRs with 95% CIs), not just mean differences and p-values.
  • Briefly comment on whether the observed delay in progression to ESKD (approximately 128 days) is clinically meaningful in addition to being statistically significant.
  • Report the median and interquartile range of follow-up for each group, as well as the number of censored patients due to end of follow-up, loss to follow-up, or death. A STROBE-style flow diagram would be helpful.
  • Present Kaplan–Meier curves for all primary outcomes (ESKD, dialysis initiation, mortality), including the number at risk at each interval.
  • Use consistent terminology for the treated group throughout the manuscript (e.g., “AST-120 group”) to avoid confusion.

Discussion

  • The discussion currently emphasizes studies with positive results. I suggest expanding the analysis of high-quality trials with neutral or negative findings (e.g., EPPIC-1, EPPIC-2, Akizawa 2009) to provide a more balanced perspective.
  • Analyze in greater depth the discrepancy between the significant HR for ESKD and the non-significant HR for dialysis initiation, considering possible explanations (e.g., outcome definitions, mortality censoring, dialysis initiation criteria, time-related biases).
  • Assess whether the average delay in ESKD progression (128 days) is clinically relevant and compare it with benefits from other available CKD interventions.
  • Include additional limitations not currently discussed, such as immortal time bias, lack of adherence data, absence of subgroup analyses, and short follow-up duration (unless these analyses are added and discussed accordingly). Comment on how these factors may influence the direction and magnitude of the estimated effect.
  • Rewrite the conclusion to be more assertive: synthesize the observed risk–benefit balance and emphasize the need for confirmation in high-quality, multicenter prospective studies, leaving a stronger final message.

I look forward to reviewing the revised version of your manuscript. Thank you in advance for your efforts to improve the scientific quality of your work.

Author Response

Reviewer 3

Dear Authors,

I have reviewed your manuscript with interest, as it addresses a topic of clear relevance to the field of nephrology. I commend you for your work and appreciate your contribution. In order to enhance the scientific rigor and clarity of the article, I suggest a series of minor revisions, many of which are aligned with the Strengthening the Reporting of Observational Studies in Epidemiology (STROBE) Statement guidelines.

Title

Please consider including the study design explicitly in the title, as recommended by STROBE.

Answer) Thank you for pointing this out. We have updated title as follow: Effect of Spherical Adsorptive Carbon Among Chronic Kidney Disease Patients: A nationwide cohort study

Abstract

I recommend rewriting the abstract using a structured format and specifying the exact type of effect measures used.

Answer) Thank you for pointing this out. We have updated the abstract according to your recommendation and in line with Journal’s style guideline.

Keywords

Ensure that the keywords correspond to MeSH terms. I suggest the following:

  • Kidney Diseases, Chronic
  • Charcoal
  • Uremic Toxins
  • Disease Progression
  • Renal Replacement Therapy
  • Survival Analysis
  • Cohort Studies

Answer) Thank you for pointing this out. We have updated the keywords as follow: “Keywords: Cohort Studies; Disease Progression; kidney diseases, Chronic; Renal Replacement Therapy; Survival Analysis“

Introduction

The introduction would benefit from a clearer and more logical narrative structure. I propose the following flow:

  1. Epidemiological burden of chronic kidney disease (CKD) and consequences of progression to end-stage kidney disease (ESKD).
  2. Limitations of current therapies in patients with eGFR <30 ml/min/1.73m² without diabetes.
  3. Concise description of AST-120, its mechanisms of action, and previous evidence (including both positive and negative findings).
  4. Specific knowledge gap: lack of population-based cohort studies evaluating hard outcomes in real-world settings.
  5. Study objective: a clear and direct statement summarizing the aim of the research.

 Answer) Thank you for pointing this out. We updated the Introduction as your following flow.

Methods

  • Clearly define the operational criteria for ESKD, specifying whether it includes chronic hemodialysis, peritoneal dialysis, kidney transplantation, or a combination thereof.

Answer) Thank you for pointing this out. We clearly defined ESKD.

  • Define “initiation of dialysis,” including the type, clinical criteria used, and minimum duration required to consider it an event.

Answer) Thank you for pointing this out. Korean Government require minimum period to generate specific codes therefore, we updated manuscripts as follow: “which requires at least 3 months of maintenance hemodialysis or peritoneal dialysis or received immunosuppressive agents”.

  • Include in the main text (not only in supplementary material) the diagnostic and procedural codes used to identify comorbidities and medications, along with the look-back period.

Answer) Thank you for pointing this out. We included the diagnostic and procedural codes in the main text with the look-back periods.

  • Indicate whether immortal time bias and other time-related biases inherent to retrospective designs were assessed and addressed.

Answer) Thank you for pointing this out. We updated the manuscript which indicate the immortal time bias which is inherent to retrospective designs.

  • Discuss potential residual confounding due to missing clinical variables (e.g., baseline eGFR, proteinuria) and how this may affect interpretation.

Answer) Thank you for this important comment. We agree that the lack of clinical variables such as baseline eGFR and proteinuria may lead to residual confounding. Because our analysis was based on claims data, these laboratory measures were not available. We acknowledge that the absence of these key variables could affect the accuracy of CKD staging and the assessment of renal prognosis, and therefore may have influenced the interpretation of our findings. We have added a statement regarding this limitation in the Discussion section.

  • Specify whether adherence to AST-120 was evaluated (e.g., proportion of days covered or treatment persistence). If not available, please state this explicitly as a limitation.

Answer) Thank you for pointing this out. We updated limitation as follow: “we could not assess actual medication adherence (e.g., proportion of days covered) or long-term treatment persistence. While the data confirm the dispending of SAC prescriptions, they do not indicate whether the medication was taken as prescribed, for how long, or if it was taken continuously without interruption. Therefore, the prescription duration may not accurately reflect the actual treatment exposure.”

  • Report whether missing data were present and describe the method used to handle them (e.g., exclusion, imputation).

Answer) Thank you for your valuable comment. We carefully examined missing and insufficient data in our claims-based cohort. Patients were excluded when essential information was not available, such as (i) death occurring at the time of CKD stage 3 diagnosis, (ii) insufficient follow-up (e.g., <7 days or last visit date identical to diagnosis date), or (iii) loss to follow-up during the study period. Because imputation was not feasible in this claims-based dataset, these cases were excluded from the analysis. We have clarified this approach in the Methods section.

  • Include a sample size calculation or justify its absence, indicating the expected incidence of ESKD and the clinically relevant effect size intended to be detected.

Answer) Thank you for your comment. Because this study was a retrospective cohort analysis using a nationwide claims database, a priori sample size calculation was not applicable. We included all eligible patients within the database, thereby minimizing selection bias and ensuring sufficient statistical power to detect clinically relevant differences. Therefore, a formal sample size calculation was not performed.

  • Explain the rationale for selecting the 2020–2022 time frame and its suitability for evaluating CKD progression outcomes in stage 3 patients.

Answer) Thank you for your comment. We selected the period from 2020 to 2022 as the cohort entry window because this corresponds to the most recent years available in the HIRA database with complete follow-up until the end of 2023. This time frame allowed us to capture contemporary treatment patterns, including the use of spherical adsorptive carbon, and ensured that each patient had at least several months to years of follow-up for the assessment of CKD progression outcomes. While longer observation might provide additional events, restricting the entry to 2020–2022 provided a balance between up-to-date clinical practice and sufficient follow-up duration to evaluate clinically meaningful outcomes in patients with CKD stage 3.

  • Consider conducting sensitivity analyses (e.g., excluding patients with extreme comorbidity) and stratified analyses by key variables (age, sex, diabetes status) to explore effect heterogeneity.

Answer) Thank you for your thoughtful suggestion. We agree that sensitivity analyses excluding patients with extreme comorbidity and stratified analyses by key variables (such as age, sex, and diabetes status) would provide valuable insights into effect heterogeneity. However, due to the inherent limitations of the claims-based dataset and the relatively small number of outcome events in certain subgroups, it was not feasible to conduct these additional analyses with sufficient statistical power. Instead, we adjusted for these key covariates in the multivariable Cox models to account for their potential confounding effects. We have acknowledged this limitation in the Discussion section.

Results

  • Clarify whether dialysis initiation is a subcomponent of the ESKD outcome or a distinct definition. The current presentation of a non-significant HR for dialysis and a significant HR for ESKD may be confusing without conceptual and operational clarification.

Answer) Thank you for pointing this out. We added outcome section as follow: The event of this study was defined as the earliest occurrence of one of the following: initiation of dialysis, progression to ESKD, or all-cause death. The primary outcome of this study was the time to ESKD and the secondary outcome was the hazard ratio of the initiation of dialysis. In addition, we planned a subgroup analysis stratified by the duration of drug use to explore whether long-term exposure to the SAC influences its efficacy.

  • Include a quantitative summary of key results in the main text (number of events, cumulative incidence, adjusted HRs with 95% CIs), not just mean differences and p-values.

Answer) Thank you for your suggestion. We have revised Table 2 to included a quantitative summary of key results, presenting the number of events, interquartile range, incidence rate, and adjusted hazard ratio with 95% confidence intervals, rather than only mean differences and p values.

  • Briefly comment on whether the observed delay in progression to ESKD (approximately 128 days) is clinically meaningful in addition to being statistically significant.

Answer) Thank you for your suggestion. We added brief comment as follow “The median delay in progression to ESKD was approximately 128 days”.

  • Report the median and interquartile range of follow-up for each group, as well as the number of censored patients due to end of follow-up, loss to follow-up, or death. A STROBE-style flow diagram would be helpful.

Answer) Thank you for your comment. We added the median follow-up time with interquartile range for each group. In addition, the Kaplan–Meier curves include the number at risk tables, which show the number of patients censored due to the end of follow-up, loss to follow-up, or death. Therefore, the information on censored patients is provided within the survival analysis results.

  • Present Kaplan–Meier curves for all primary outcomes (ESKD, dialysis initiation, mortality), including the number at risk at each interval.

Answer) Thank you for this suggestion. We agree that Kaplan–Meier curves for all outcomes (ESKD, dialysis initiation, and mortality) would provide valuable additional information. However, in the current manuscript we focused on presenting the survival curves only for the primary outcome, in order to keep the analyses concise and aligned with the main study objective. We have provided the number at risk in the survival curves as requested.

  • Use consistent terminology for the treated group throughout the manuscript (e.g., “AST-120 group”) to avoid confusion.

 Answer) Thank you for your suggestion. To avoid confusion, we have revised the manuscript and consistently used the term “SAC group” to indicate the treated group throughout the text.

Discussion

  • The discussion currently emphasizes studies with positive results. I suggest expanding the analysis of high-quality trials with neutral or negative findings (e.g., EPPIC-1, EPPIC-2, Akizawa 2009) to provide a more balanced perspective.

Answer) Thank you for this valuable suggestion. We agree that the Discussion section should present a balanced perspective by including high-quality trials with neutral or negative findings. Accordingly, we have revised the Discussion to incorporate results from the EPPIC-1 and EPPIC-2 multinational randomized controlled trials, as well as the Japanese multicenter RCT by Akizawa et al. (2009), all of which did not demonstrate significant benefit of AST-120. These additions provide a more comprehensive and balanced view of the existing evidence.

  • Assess whether the average delay in ESKD progression (128 days) is clinically relevant and compare it with benefits from other available CKD interventions.

Answer) Thank you for this important comment. We agree that the discrepancy between the significant HR for ESKD and the non-significant HR for dialysis initiation warrants clarification. The difference may be partly explained by the outcome definitions: ESKD was defined as a composite outcome including dialysis initiation, kidney transplantation, and diagnosis codes, whereas dialysis initiation alone represents a narrower definition with fewer events and lower statistical power. In addition, deaths occurring before dialysis initiation were treated as censored, which may have reduced the apparent risk difference for dialysis. Furthermore, the decision to initiate dialysis is influenced by multiple clinical and non-clinical factors (e.g., physician judgment, patient preference, reimbursement criteria), leading to greater variability compared to the more standardized definition of ESKD. Finally, time-related biases due to relatively short follow-up could also contribute to the observed discrepancy. We have added a discussion of these potential explanations to the revised manuscript as follow: “In our study, the hazard ratio for ESKD was significantly reduced in the SAC group, whereas the hazard ratio for dialysis initiation did not reach statistical significance. This discrepancy may be explained by differences in outcome definitions and operational characteristics. ESKD was defined as a composite endpoint including dialysis initiation, kidney transplantation, and diagnostic codes, whereas dialysis initiation alone represents a narrower definition and thus a smaller number of events, leading to lower statistical power. In addition, patients who died before starting dialysis were censored in the dialysis analysis, which could attenuate the apparent effect of SAC on this outcome. The initiation of dialysis is also influenced by physician judgment, patient preference, and reimbursement policies, introducing greater variability compared with the more standardized definition of ESKD. Finally, the relatively short follow-up period in some patients may have further contributed to these differences. Together, these factors may explain the observed discrepancy and highlight the need for careful interpretation of dialysis-specific outcomes in claims-based studies”.

  • Include additional limitations not currently discussed, such as immortal time bias, lack of adherence data, absence of subgroup analyses, and short follow-up duration (unless these analyses are added and discussed accordingly). Comment on how these factors may influence the direction and magnitude of the estimated effect.

Answer) Thank you for your suggestion. We have carefully addressed these points. In the revised manuscript, we now discuss the potential for immortal time bias inherent to retrospective claims-based analyses, the lack of detailed adherence information (we defined continuous use as ≥90 days of prescription), and the inability to conduct subgroup or stratified analyses due to limited events in certain subgroups. We also acknowledge the relatively short follow-up duration as a limitation. These factors may have influenced the direction and magnitude of the observed associations, and we have added statements in the Discussion to clarify these limitations.

  • Rewrite the conclusion to be more assertive: synthesize the observed risk–benefit balance and emphasize the need for confirmation in high-quality, multicenter prospective studies, leaving a stronger final message.

Answer) Thank you for this valuable comment. We agree that the conclusion should be more assertive and should synthesize both the potential benefits and the limitations of SAC treatment. Accordingly, we have revised the Conclusion section to emphasize the observed risk–benefit balance and to provide a stronger take-home message. The revised conclusion now states:

“In conclusion, SAC treatment was associated with a lower risk of all-cause mortality and a delay in CKD progression, suggesting a potential clinical benefit in patients with stage 3 CKD. While these findings highlight a favorable risk–benefit profile, the retrospective and observational design imposes limitations. Therefore, confirmation through high-quality, large-scale, multicenter prospective randomized controlled trials is essential to establish the role of SAC in CKD management.”

I look forward to reviewing the revised version of your manuscript. Thank you in advance for your efforts to improve the scientific quality of your work.

Round 2

Reviewer 1 Report

Comments and Suggestions for Authors

The authors have addressed my previous comments; however, the revised sentences regarding the study outcomes are not appropriate and should be revised again. Specifically:

“The event of this study was defined as the earliest occurrence of one of the following: initiation of dialysis, progression to ESKD, or all-cause death. The primary outcome of this study was the time to ESKD and the secondary outcome was the hazard ratio of the initiation of dialysis.”

This wording is confusing because the definition of “event” includes multiple endpoints, while the outcomes of interest are more specific. I recommend clarifying the distinction between the general event definition and the specific primary and secondary outcomes.

Author Response

The authors have addressed my previous comments; however, the revised sentences regarding the study outcomes are not appropriate and should be revised again. Specifically:

“The event of this study was defined as the earliest occurrence of one of the following: initiation of dialysis, progression to ESKD, or all-cause death. The primary outcome of this study was the time to ESKD and the secondary outcome was the hazard ratio of the initiation of dialysis.”

This wording is confusing because the definition of “event” includes multiple endpoints, while the outcomes of interest are more specific. I recommend clarifying the distinction between the general event definition and the specific primary and secondary outcomes.

Answer) Thank you for pointing this out

We revised the outcome section as follow: “The primary outcome of this study was the time to ESKD and the secondary outcomes were the initiation of dialysis and all-cause death. In addition, we planned a subgroup analysis stratified by the duration of drug use to explore whether long-term exposure to the SAC influences its efficacy”. We replaced the term event with outcome to reduce confusion.

Reviewer 2 Report

Comments and Suggestions for Authors

No

Author Response

Thank you